

# Deterministic processes dominate soil microbial community assembly in subalpine coniferous forests on the Loess Plateau

Pengyu Zhao[1], Jiabing Bao[1], Xue Wang[1], Yi Liu[2], Cui Li[2] and Baofeng Chai[1]

[1] Institute of Loess Plateau, Shanxi University; Shanxi Key laboratory of Ecological Restoration of Loess Plateau, Taiyuan, China

[2] College of Resources and Environment, Shanxi University of Finance and Economics, Taiyuan, China

## ABSTRACT

Microbial community assembly is influenced by a continuum (actually the trade-off) between deterministic and stochastic processes. An understanding of this ecological continuum is of great significance for drawing inferences about the effects of community assembly processes on microbial community structure and function. Here, we investigated the driving forces of soil microbial community assembly in three different environmental contexts located on subalpine coniferous forests of the Loess Plateau in Shanxi, China. The variation in null deviations and phylogenetic analysis showed that a continuum existed between deterministic and stochastic processes in shaping the microbial community structure, but deterministic processes prevailed. By integrating the results of redundancy analysis (RDA), multiple regression tree (MRT) analysis and correlation analysis, we found that soil organic carbon (SOC) was the main driver of the community structure and diversity patterns. In addition, we also found that SOC had a great influence on the community assembly processes. In conclusion, our results show that deterministic processes always dominated assembly processes in shaping bacterial community structure along the three habitat contexts.

## INTRODUCTION

Understanding the fundamental ecological mechanisms that drive the assembly processes of microbial communities is a major challenge in community ecology (*Shen et al., 2013*), particularly microbial ecology. The assembly processes of the microbial community in a local community are generally influenced by two types of ecological processes, including deterministic and stochastic processes. First, deterministic factors, such as organism traits, interspecies relationships (e.g., competition, predation, mutualisms, and trade-offs), and environmental factors (e.g., pH, temperature, salt, and moisture) govern the community structure (*Chase & Myers, 2011*; *Dumbrell et al., 2010*; *Ofiţeru et al., 2010*). Ecologists have traditionally appreciated that the environmental context determines the assembly processes of microbial communities: "Everything is everywhere, but the environment selects" (*Baas*

Corresponding author
Baofeng Chai, bfchai@sxu.edu.cn

*Becking, 1934*). For example, environmental factors such as pH (*Tripathi et al., 2018*), temperature (*Anderson & Laurel, 2013*), or nitrogen levels (*Xiong et al., 2016*) may be major determinants of microbial community structure.

For the other type of community assembly processes (i.e., stochastic processes), it is assumed that community structures are independent of organism traits and are governed by birth, death, colonization, extinction, drift, and speciation (*Hubbell & Borda-de-Água, 2004*), and it is hypothesized that species are all ecologically equivalent (*Woodcock et al., 2007*). Previous studies have confirmed that both deterministic and stochastic processes act concurrently to regulate the assembly of ecological communities (*Diniandreote et al., 2016*; *Diniandreote et al., 2015*; *Zhou & Ning, 2017*), but the relative importance may vary in different environmental contexts (*Tian et al., 2017*). This may be because the variation in ecological selection strength and the rates of dispersal on different habitat contexts can influence the relative importance of deterministic and stochastic processes across temporal and spatial scales, in addition to within entire ecosystems (*Chisholm & Pacala, 2011*; *Jurburg et al., 2017*). Therefore, investigation into community driving forces in different habitats can enrich the understanding on the community assembly process.

In this study, soil was sampled from 23 soil plots in subalpine coniferous forests located on the Loess Plateau in Shanxi province, China. The 16S ribosomal RNA genes of bacteria were analyzed using high-throughput sequencing. To investigate the driving forces of soil microbial community assembly, we sampled three sites having different environmental characteristics. Sampling was performed along three different altitudinal gradients. This study can largely enrich the understanding on microbiology of subalpine mountains. Our aims were as follows: (i) to quantify the relative roles of deterministic and stochastic processes in bacterial community dynamics for three different habitat contexts; and more precisely (ii) to evaluate the effects of environmental factors on microbial community assembly.

## MATERIAL AND METHODS

### Site and sampling

A total of 23 soil plots were sampled (Table S1, Fig. S1) in August 2016 and August 2017. The sites were selected because their vegetation was subalpine mountain coniferous forests and they were located between 1,900 m and 3,055 m above mean sea level (amsl). The study area has a warm temperate continental monsoon climate, and mostly cinnamon soil.

This study focused on response patterns along environmental gradients rather than exploring differences among treatment groups. Thus, we sampled along three altitudinal gradients without replicates. Previous studies have shown that for continuous environmental drivers, gradient designs further allow for better extrapolation, characterization of (nonlinear) response functions, and, consequently, quantitative outputs better suited for ecological models than replicated designs (*Cottingham, Lennon & Brown, 2005*).

To avoid the interference of vegetation factors, we sampled plots in the single vegetation type (i.e., *Larix principis-rupprechtii* forests). These sites located on subalpine

ecological environments possess pronounced climatic gradients and climosequences within short distances, with a high level of environmental heterogeneity (*Siles & Margesin, 2017*). Therefore, the sites with different altitudinal gradients corresponded to different environmental contexts and different environment characteristics.

Eight plots were sampled from the Wutai Mountain site (WT), which ranges from 1,900 m and 3,055 m amsl. Ten plots were sampled from the Pangquangou Natural Reserve site (PQG), ranging from 1,950 m to 2,650 m amsl. The last, five plots were sampled from the Luya Mountain site (LY), which ranges from 2,000 m and 2,400 m amsl. The details of each sample plots were added in the Supplemental Files (Table S1). At each sampling site, a 1 m × 1 m sampling plot was established *in situ* along the elevation gradient. Five soil cores at a depth of 15 cm were taken at each sampling plot, and then combined to form a single independent soil sample. Then, the soil samples were sealed in plastic bags and refrigerated, immediately transported to the laboratory and sieved using a 2 mm mesh. The soil samples were then stored at −80 °C until further analysis.

The soil samples were subsampled for molecular analysis and the DNA from of 1 g of soil was extracted using an E.Z.N.A.@ Soil DNA Kit (Omega Bio-tek, Inc., Norcross, GA, USA). The quality and quantity of the DNA extracts were measured using an Infinite 200 PRO plate reader (TECAN, Männedorf, Switzerland). The DNA purity was assessed based on the A260/A280 absorbance ratios, and only DNA extracts with absorbance ratios of 1.8~2.0 were used for further analyses. Three DNA samples were extracted, from each soil sample, which were then combined and sequenced at Shanghai Personal Biotechnology Co., Ltd. on an Illumina MiSeq sequencing platform based on the bacterial v3–v4 hypervariable region using bacterial 16S universal primers (341F 5′-ACTCCTACGAGGAGCA- 3′ and 805R 5′-TTACCGCGGCTGCTGGCAC- 3′) (*Tripathi et al., 2018*).

## Bioinformatics analysis

The sequencing data were analyzed using the QIIME pipeline (v1.8.0, http://qiime.org/) (*Caporaso et al., 2010*). The filtered sequence alignments were denoised by DeNoiser (*Reeder & Knight, 2010*) and then screened for chimeras using UCHIME (*Edgar et al., 2011*). The Archaea and unknown sequences were removed. The sequences were clustered into operational taxonomic units (OTUs) at a 97% similarity level using the average neighbor method and taxonomy was blast to SILVA database by k-mer searching using MOTHUR (*Pruesse et al., 2007*). The OTU table was rarefied to 4,020 sequences per sample. Ten independent maximum-likelihood phylogenetic trees based on Jukes–Cantor distance were then constructed using FastTree2 (*Price, Dehal & Arkin, 2009*) after the removal of gaps and hypervariable regions using a Lane mask supplied by QIIME to support phylogenetic diversity calculations.

## Environmental variables

In the laboratory, soil total carbon (TC), total nitrogen (TN), and total sulfur (TS) were measured using an elemental analyzer (Vario EL/ MACRO cube, Elementar, Hanau, Germany); nitrate nitrogen ($NO_3^-\_N$), ammonium nitrogen ($NH_4^+\_N$), and nitrite nitrogen ($NO_2^-\_N$) were measured by an Automated Discrete Analysis Instrument

(CleverChem 380; DeChem-Tech, Hamburg, Germany). After shaking the soil: water suspension (1:2.5 mass/volume) for 30 mins, the soil pH was measured using a pH meter (Hl 3221, Italy). The soil organic carbon in each soil sample was measured using the potassium dichromate volumetric method (*Nelson & Sommers, 1982*).

## Null model analysis

A null model was constructed to account for changes in β-diversity while controlling for stochastic variation and associated changes in α-diversity (i.e., local species richness; 999 iterations) (*Chase et al., 2011*). We considered the null deviation as the relative difference between the observed β-diversity and the null-model β-diversity (*Tucker et al., 2016*). As such, null deviation values may represent communities that are more similar than expected by chance (a negative null deviation value), less similar than expected by chance (a positive null deviation value), or close to the chance expectation (values near zero) (*Tucker et al., 2016*).

## Phylogenetic analysis

Our study used phylogenetic turnover between communities to infer ecological processes (*Stegen et al., 2015*). To quantify phylogenetic turnover between communities, we used the between community mean-nearest-taxon-distance (βMNTD) metric. βMNTD was calculated in R (*R Core Team, 2018*) 'comdistnt' (abundance.weighted = TRUE; package "picante"). Then, we evaluated β-Nearest Taxon Index (βNTI), which expresses the difference between observed βMNTD and the mean of the null distribution in units of standard deviations (*Stegen et al., 2013*).

In addition, to distinguish more details in the assembly processes, we used the Raup–Crick metric (*Chase et al., 2011*), extended to incorporate species' relative abundances; referred to as $RC_{bray}$. The R script of $RC_{bray}$ can be found at https://github.com/stegen/Stegen_etal_ISME_2013.

In a given community, we estimated the relative influence of variable selection or homogeneous selection as the fraction of their comparisons with $\beta NTI > +2$ or $\beta NTI < -2$, respectively. We regard the fraction of the between community comparisons with $|\beta NTI| < 2$ and $RC_{bray} > +0.95$ as dispersal limitation, while $|\beta NTI| < 2$ and $RC_{bray} < -0.95$ is considered homogenizing dispersal (*Diniandreote et al., 2015*; *Stegen et al., 2013*; *Stegen et al., 2015*).

## Network analysis

The co-occurrence network was constructed based on the Spearman correlation matrix offered in the 'psych' package in R. In this network, the nodes represent OTUs and the edges that connect these nodes represent correlations between OTUs. Only those connections with correlation coefficients >0.6 and $P < 0.05$ were used in the network. Thus, positive correlations indicate co-occurring OTUs based on abundances, whereas negative correlations indicate that the OTUs are mutually exclusive (*Barberán et al., 2012*). $P$-values were false discovery rate (FDR) adjusted to control for the analysis (FDR < 0.05). The network analysis was completed using the 'igraph' package in R.

## Statistical analysis

All statistical analyses were performed in the R environment using the 'vegan', 'ggplot2', 'ggpubr', and 'corrplot' packages. A Venn diagram was used to visualize the shared OTUs among the sites. A correlation matrix graph was used to demonstrate the correlation between soil physicochemical factors and was constructed using the 'corrplot' packages in R. Multivariate regression tree analysis (MRT) was used to explain the relationship between bacterial $\alpha$-diversity estimates and environmental variables in a visualized tree, and diversity indices were normalized to the same mean before performing MRT analysis (*Ge et al., 2008*). Based on the longest gradient lengths from the results of detrended correspondence analysis (DCA), we selected redundancy analysis (RDA) to quantify the effects of environmental variables on microbial community composition (*Mo et al., 2018*). Forward selection of PCNM variables based on permutation tests was chosen to identify two of the 23 extracted PCNM variables that significantly ($P < 0.05$) explained the spatial structure. The PCNM eigenfunctions, which represent the 'spectral decomposition of the spatial relationship across sampling locations', can be considered as the spatial variables in the ordination-based analysis. The contributions of environmental filtering and the space variable (PCNM) to the variation in bacterial community composition were calculated by using variance partitioning analysis (VPA) (CANOCO for Windows Version 5.0). The mantel test was performed in the R environment using the 'vegan' packages.

# RESULTS

## Physicochemical properties of the soils from the different sites

The soil physicochemical properties varied across the different sampling sites (Fig. 1). Briefly, the contents of ammonium nitrogen and nitrite nitrogen were the highest at LY sites (36.91 and 0.16 mg kg$^{-1}$, respectively), and were lowest at WT sites (17.41 and 0.04 mg kg$^{-1}$, respectively). The contents of nitrate nitrogen (6.45 mg kg$^{-1}$), SOC (70.29 mg g$^{-1}$), TC (6.4%), and TN (0.51%) were the highest at WT sites, and were the lowest at LY sites.

TN was significantly positively correlated with TC and SOC ($P < 0.05$) and significantly negatively correlated with pH value ($P < 0.05$; Fig. 2). TC and pH showed a significant negative correlation ($P < 0.05$). SOC was significantly positively correlated with nitrate nitrogen ($P < 0.05$) and significantly negatively correlated with nitrite nitrogen ($P < 0.05$). This indicated that the sites sampled had different environment characteristics.

## Dynamics of bacterial community composition and diversity

A total of 4,258 OTUs were identified from 1,062,241 high-quality sequences recovered from 23 soil samples. Good's coverage ranged from 95.19% to 99.75%, indicating that the identified sequences were representative of most of the bacterial sequences in the collected soil samples. Rarefaction curve analyses, which generally yielded asymptotic curves, indicated that the numbers of sampling plots were enough. Detailed information of the sequencing results is provided in Table S2.

The soil microbial community composition varied across the different sampling sites (Fig. 3). There were 15 bacterial phyla with relative abundances of more than 0.01%

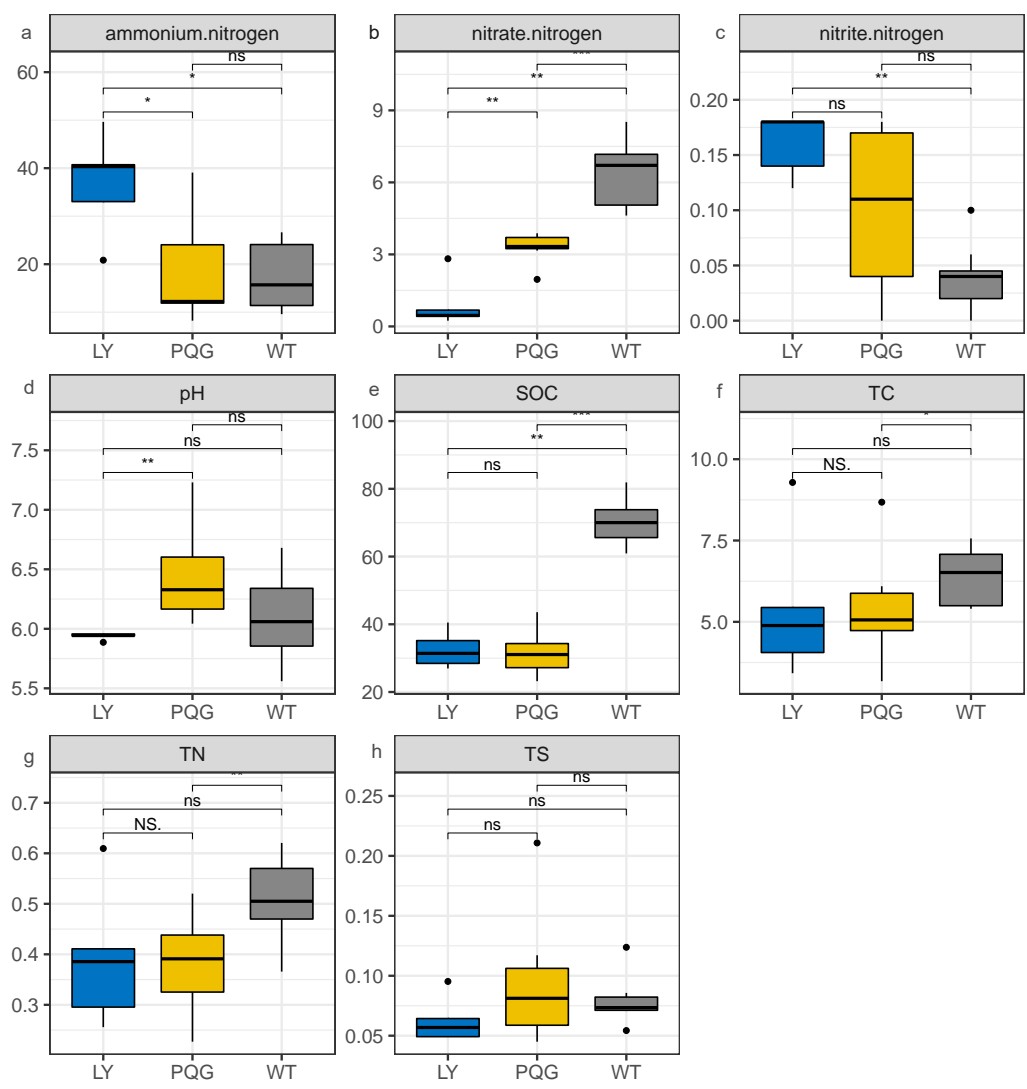

**Figure 1** **Bar plots indicating the soil physicochemical factors at different sites.** (A) Ammonium nitrogen; (B) nitrate nitrogen; (C) nitrite nitrogen; (D) pH; (E) SOC; (F) TC; (G) TN; (H) TS.

(Fig. 3A). As shown in the Venn diagram, 869 bacterial shared OTUs were observed in all sampling sites. There were 46 bacterial phyla identified (Fig. 3B). The abundance of *Proteobacteria* at all sites was the highest (mean relative abundance = 30.59%), and followed by *Acidobacteria* (19.63%), *Actinobacteria* (16.51%) and *Chloroflexi* (13.22%). Briefly, the mean relative abundance of *Proteobacteria* was the most at PQG (34.39%), and that of *Actinobacteria* was the highest at LY (26.29%). The mean relative abundances of *Acidobacteria* (28.68%) *and Chloroflexi* (16.09%) were the highest at WT. There were 31 bacterial families with relative abundances of more than 0.01% (Fig. 3C). Based on the clustering graph, the sampling plots of each of the sites roughly clustered together (Fig. 3D). The community $\alpha$-diversity indices varied at the different sites (Fig. 4). Briefly, the phylogenetic diversity (pd) and the number of observed species (sobs) were the highest
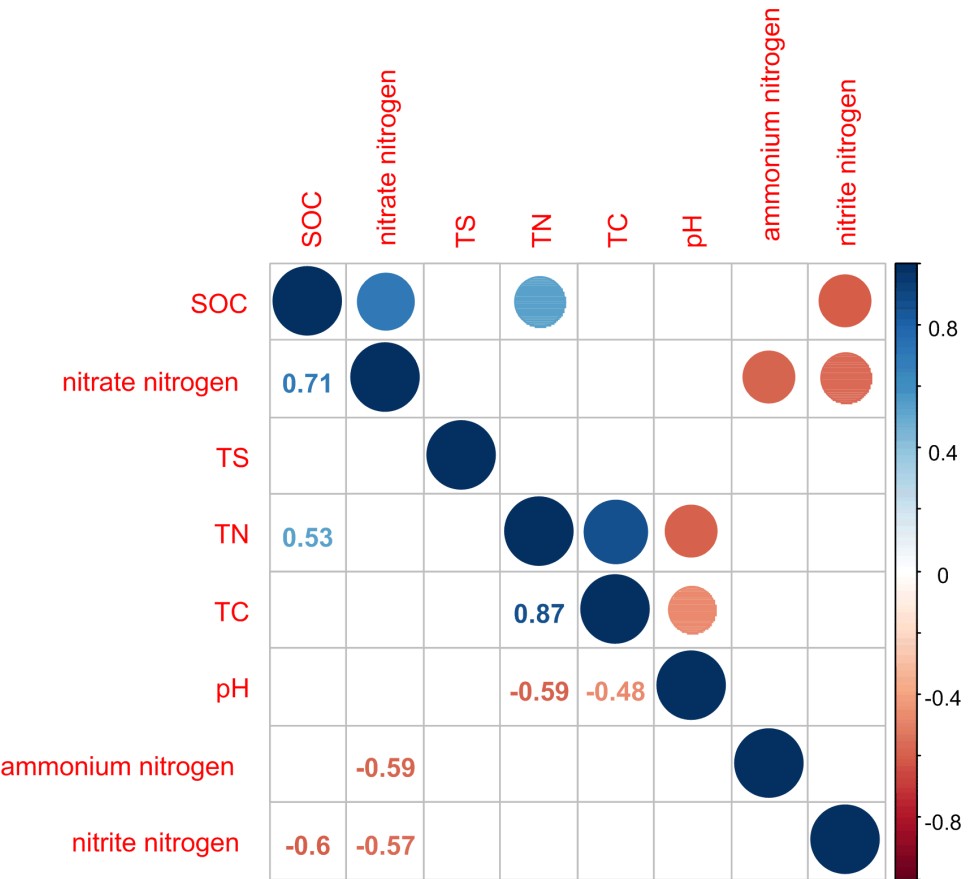

**Figure 2 Correlation matrix graph indicating the correlation between soil physicochemical factors.** Only the environmental factors with significantly difference represented in the figure.

at WT sites ($P < 0.05$). There was no significant difference in the ACE index, Chao index, Shannon index and Simpson index at the different sites ($P > 0.05$). This indicated that the sites sampled had different soil microbial community structure.

## Effects of environmental factors on microbiome dynamics

Based on the results of the DCA (axis length = 1.02), we used RDA to identify the abiotic environmental drivers that influenced bacterial community composition (Fig. 5; permutation test, $P < 0.01$). The results demonstrated that *Proteobacteria, Bacteroidetes,* and *Cyanobacteria* were mainly driven by pH, while SOC, TC, and TN were the main abiotic drivers of *Parcubacteria* and *Planctomycetes*.

In the MRT analysis (Fig. 6), we observed that the diversity indices (normalized) were mainly split by SOC, explaining 36.75% in the first spilt. The correlation analysis showed similar results: SOC was significantly correlated with bacterial communities at the phylum level (e.g., *Proteobacteria, Bacteroidetes,* and *Chloroflexi*). Given its contribution to explaining community distribution patterns, SOC was further used as a descriptor for the environmental gradients.

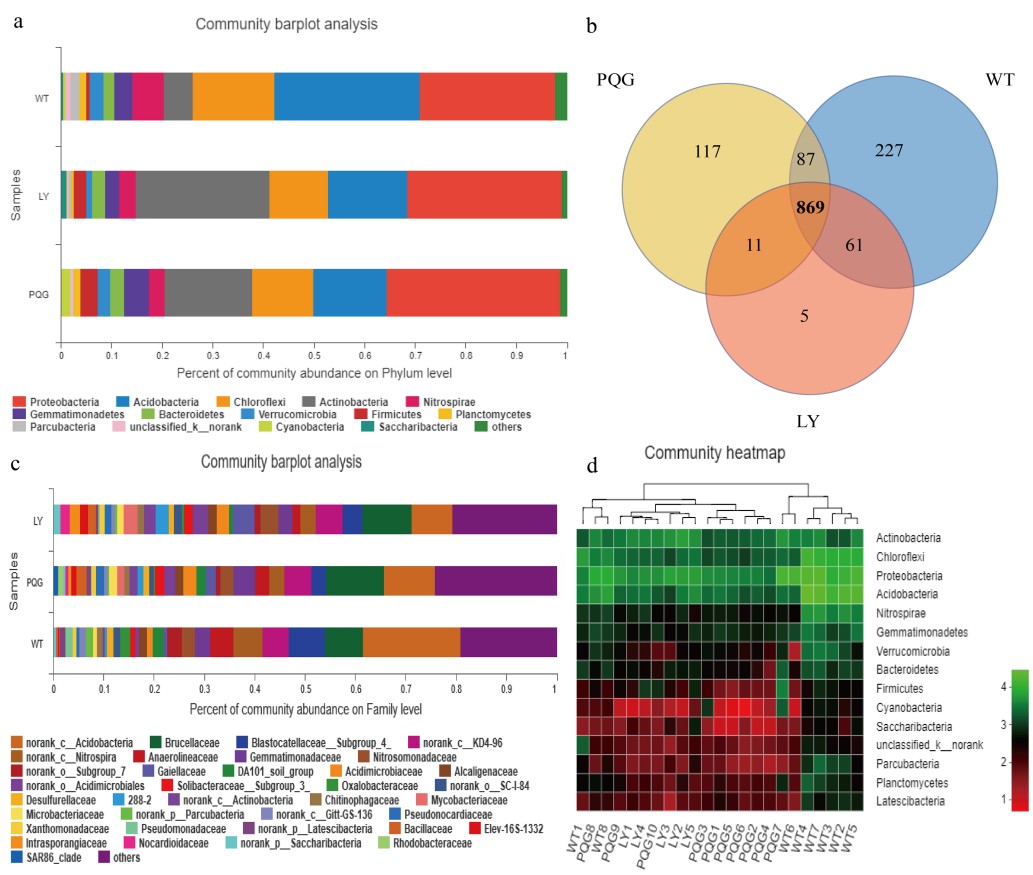

**Figure 3 Microbial community composition and structure.** Relative abundance of the dominant bacterial phyla (A) and family levels (C) across the sites. Venn diagram (B) showing the shared OTUs in all plots. In the heat map (D), the horizontal coordinate represents the sample name, and the vertical coordinate represents the species name. A color gradient is used to represent the proportion of species. The value on each site represent average values of sampling plots.

The variation partitioning analysis showed that environmental variables (20.3%) explained more variation of microbial community structure than spatial variables (1.9%). This suggested that both deterministic and stochastic processes were involved in the assembly of microbial communities, and that deterministic processes were dominant. The unexplained variable was 78.6% (Fig. 7).

## Nonrandom co-occurrence patterns of the microbial community

Network analysis was applied to explore the interspecific relationship patterns in the microbial communities (Fig. 8). Compared with the LY- and WT- network, the PQG-network exhibited more edges (87), more vertices (40), more modularity (0.691), higher average degree (4.35) and average clustering coefficients (0.858), but less the numbers of modules (6) (Table S3). Strong positive correlations were observed at all sites, while negative correlations were rare. The size of the nodes corresponds to betweenness centralization values.

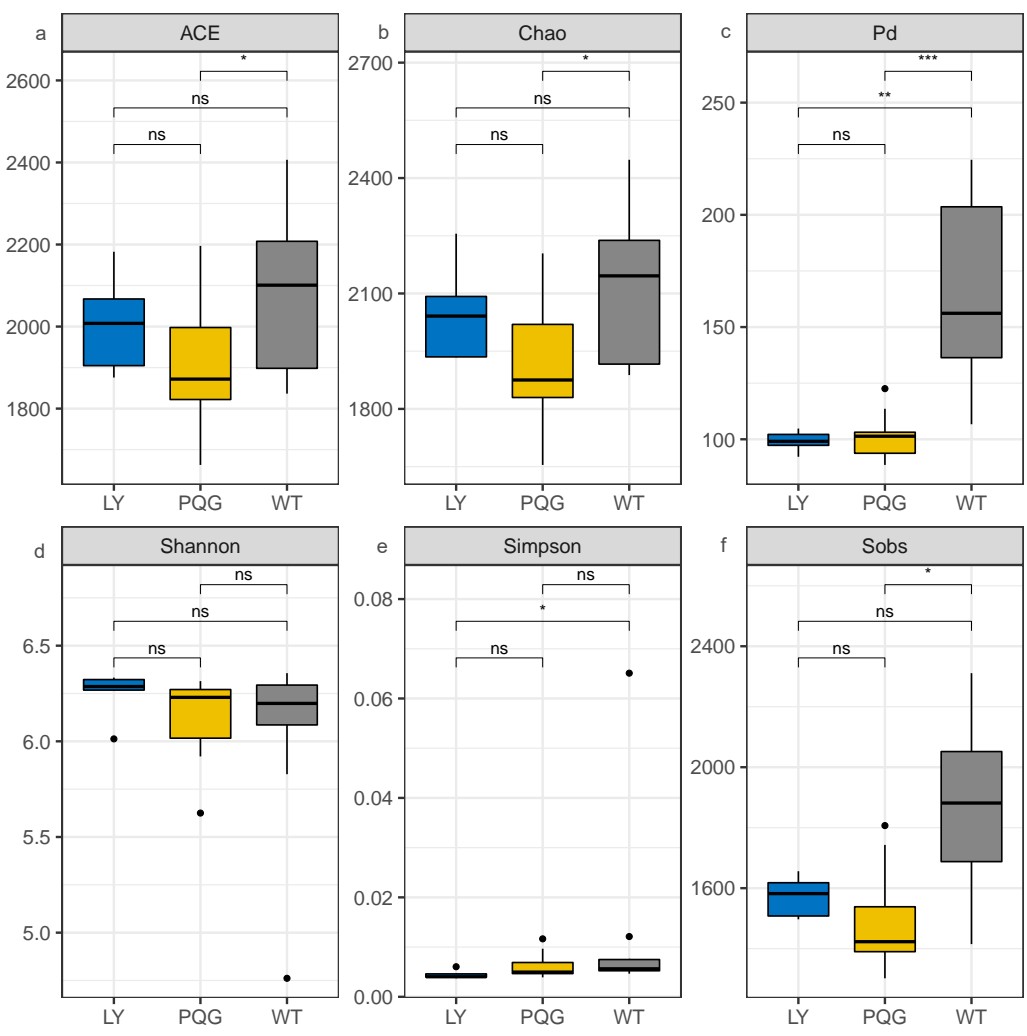

**Figure 4 Bacterial community diversity at the different sites.** (A) ACE; (B) Chao; (C) Pd; (D) Shannon; (E) Simpson; (F) Sobs.

## The bacterial community assembly processes

According to the null model analysis, our results demonstrated that the null deviation values varied at different sites (ranging from 0.29 to 0.57; Fig. 9A). The bacterial communities at WT deviated significantly from the null expected value (relative null deviation = 0.45) and were greater than that at LY site and PQG site (relative null deviation = 0.32 and 0.34, respectively) ($P < 0.05$).

Most importantly, we observed that the microbial community was more greatly shaped by variable selection ($\beta$NTI > +2) (Fig. 9B). From LY to WT, we observed a gradual increase in the relative role of deterministic processes compared to stochastic processes (Fig. 9C). Based on the regression analysis of the environmental variables with assembly process parameters, we found that SOC had a great influence on community assembly
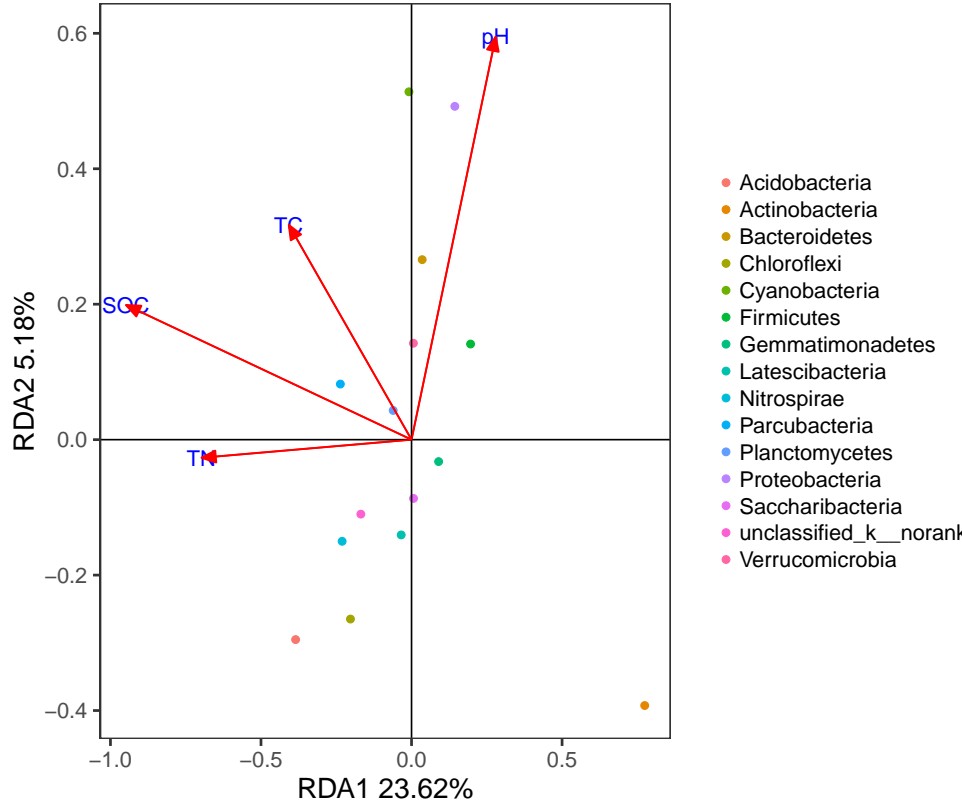

**Figure 5** RDA of the bacterial communities and the response of these communities to significant soil physicochemical properties.

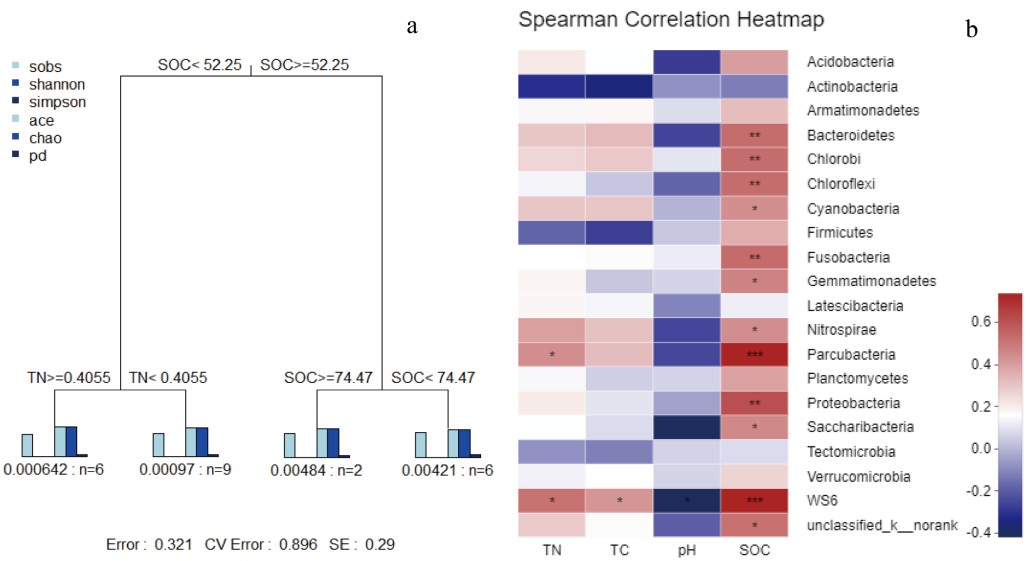

**Figure 6** MRT of bacterial α-diversity data associated with key environmental factors (A); correlation analysis (B) based on spearman correlation of microbial community composition and soil physico-chemical factors.

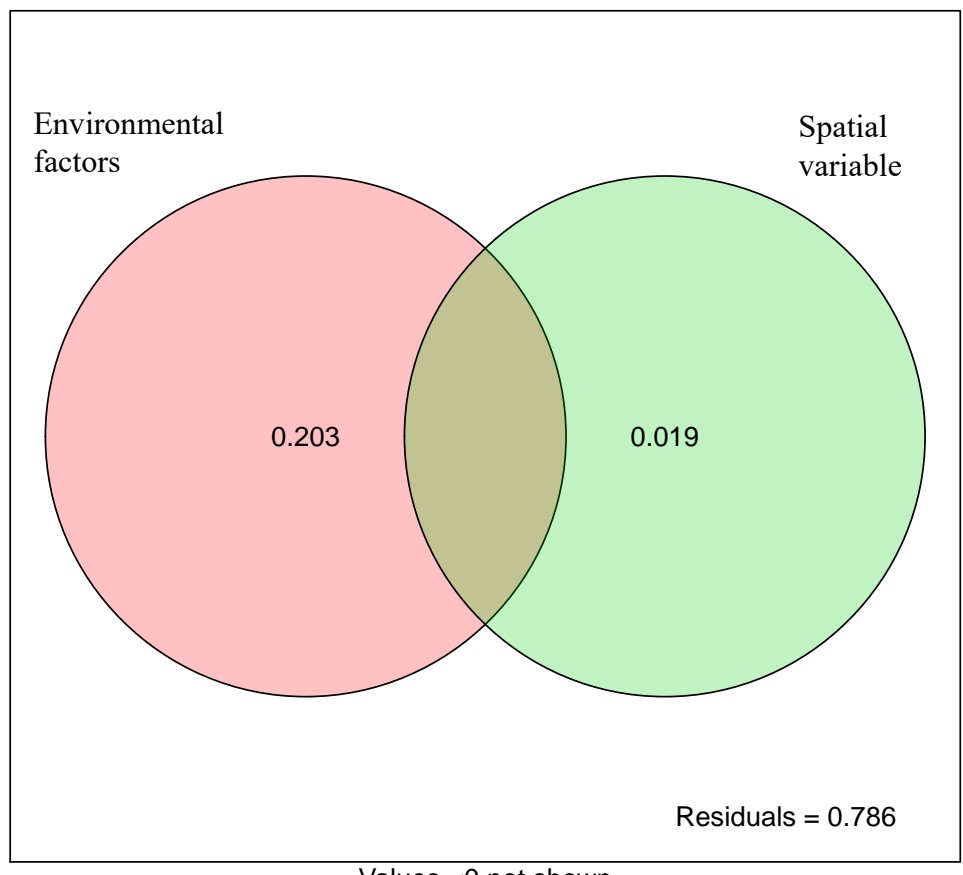

**Figure 7** Variation partitioning analysis showing the percentages of variance in bacterial communities explained by environment factors and spatial variable (PCNM).

processes (Fig. 9D). The mantel test between βNTI and SOC matrices indicated the similar conclusion ($P < 0.05$, $R = 0.509$).

## DISCUSSION

Compared to LY, the microbial community at WT was more greatly driven by deterministic processes. The driving effects of the deterministic processes gradually increased from LY to WT. Given this, we inferred that a continuum existed between deterministic and stochastic processes in the assembly of microbial communities in the study area. This is consistent with previous studies (*Chase et al., 2011*; *Tucker et al., 2016*; *Jurburg et al., 2017*; *Tian et al., 2017*), which pointed out the relative importance of the two processes varied in the different environmental contexts. For example, in terms of plants, aggregation in temperate forests reflect stronger environmental correlations, suggesting a key role for species-sorting processes (deterministic processes) (*Myers et al., 2013*). In terms of microorganisms, previous studies have noted that bacterial community assembly is largely governed by stochastic processes in early successional soils, with the relative roles of

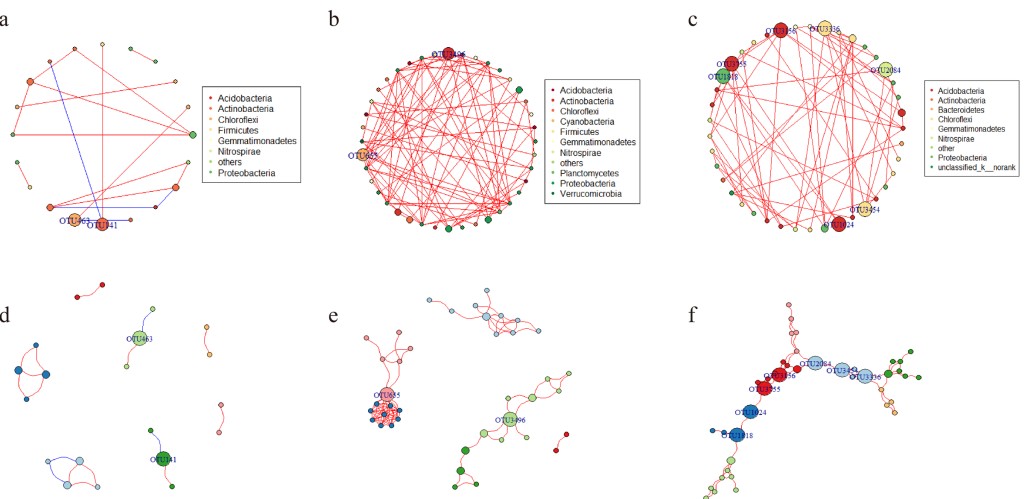

**Figure 8  Network of co-occurring OTUs.** A and D represent the network of the microbial community for LY; B and E for PQG; C and F for WT. Furthermore, A, B and C represent the network analysis colored by phylum, where D, E and F represent the network analysis colored by modular class. A red line indicates a positive interaction between two individual nodes, while a blue line indicates a negative interaction. The size of the nodes corresponds to betweenness centralization values.

deterministic processes increasing progressively in later successional soils (*Diniandreote et al., 2015*; *Ferrenberg et al., 2013*; *Hanson et al., 2012*).

Previous research has confirmed this continuum could be dependent on varying environmental conditions and the characteristics of organisms (*Zhou et al., 2013*). Environmental factors, such as salinity (*Lozupone & Knight, 2007*), pH (*Fierer & Jackson, 2006*; *Griffiths et al., 2011*), C/N ratio (*Bates et al., 2011*), soil C (*Drenovsky et al., 2004*), nitrogen levels (*Xiong et al., 2014*), and the structure of the plant community (*Lundberg et al., 2012*) may be major determinants of microbial community structure. Our results demonstrated that pH, SOC, TC, and TN were the main abiotic drivers of microbial community compositions. More importantly, based on the integrated results of the MRT analysis, RDA, and correlation analysis, we identified SOC as a general descriptor that encompassed the environmental gradients by which the communities responded to.

Our results demonstrated that SOC differed significantly at different sites, and was significantly correlated with nitrate nitrogen, nitrite nitrogen and TN ($P < 0.05$). This indicated that SOC was closely related to soil fertility and possessed the highest weighting. Litters from the trees will impact SOC, which in turn will impact the community assembly structure, and this is perhaps the reason explaining why variable selection increases from LY to WT sites. The relationships between SOC and bacterial community assembly have also been reported across a broad range of microbial ecosystems (*Bastida et al., 2013*). Most importantly, we also observed that SOC was closely associated with the community assembly process. Similar results reported that the relative roles of stochastic and deterministic processes can vary with the successional age of soils and can primarily be attributed to the covariance of soil pH with age (*Tripathi et al., 2018*). The unexplained variation in VPA

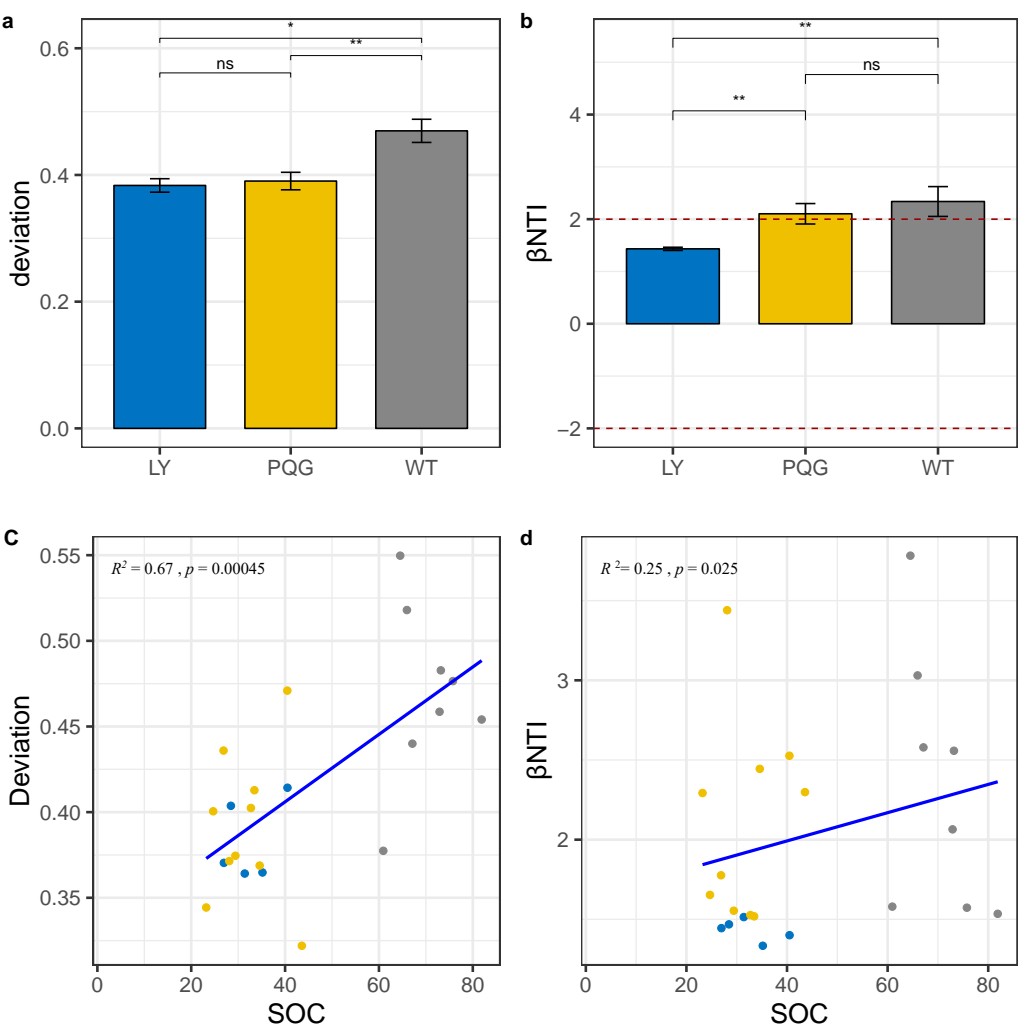

**Figure 9 Microbial community assembly processes.** The $\beta$-diversity null model analysis showing the null deviation of the bacterial communities at different sites (A). A null deviation close to zero suggests that stochastic processes are more important in structuring the community, whereas larger positive or negative null deviations suggest that deterministic processes play more important roles. Bar plot indicates that $\beta$NTI values varied among sites, but were all greater than +2 (B). Regression analysis of the environmental variables based on the results of the assembly processes parameters (C, D). We used the analysis of variance (ANOVA) to evaluate differences in the different indices. ns, not significantly; * $0.01 < P \leq 0.05$; ** $0.001 < P \leq 0.01$; *** $P \leq 0.001$.

(78.6%) could be due to stochastic influences (e.g., drift or speciation *Caruso et al., 2011*), unmeasured soil physicochemical properties (e.g., metal ion concentration *Gombeer et al., 2015*) or interactions between species (e.g., competition *Caruso et al., 2011*). In fact, in other studies of microbial communities using VPA, the unexplained portions may also account for more than 50% (*Liao et al., 2016*; *Mo et al., 2018*).

In deterministic processes, not only environmental filtering, but also interspecies interactions have a great influence on community assembly. Ecologists recently accepted that competition and environmental processes act simultaneously (*Zhang et*

*al., 2018*). In the network analysis, the higher modularity indicates that the network became denser, suggesting that the microbial communities are highly complex (*Olesen et al., 2007*). Interestingly, the modularity was the highest at PQG (0.691). This may be related to the greater sampling scales and elevation gradients, and thus greater environmental heterogeneity at PQG. The average path distance represents the shortest path between two nodes (*Wang et al., 2016*), which demonstrated irregular variation at WT (*Zheng et al., 2017*). Strong positive correlations were observed among sites, while negative correlations were rare (Figs. 8A–8C). This implied that microbes might cooperate in order to adapt to similar niches. In the network, positive links could be attributed to niche overlap and cross-feeding, while negative relationships could be attributed to competition and amensalism (*Faust & Raes, 2012*). From an ecological perspective, the peripherals may represent specialists, whereas module hubs and connectors may be more generalists and network hubs may be super-generalists (Figs. 8D–8F) (*Deng et al., 2012*). It is interesting to observe that the module hubs and connectors differed at the different sites.

## CONCLUSION

We quantified the importance of the deterministic and stochastic processes driving the bacterial community assembly on different sites in subalpine coniferous forests, and showed that deterministic processes prevailed. Moreover, SOC was closely related to microbial community structure and greatly influenced the processes of community assembly.

## ACKNOWLEDGEMENTS

We are grateful to all the scientists who contributed to the collection of data used in this study.

### Funding

This work was supported by the National Natural Science Foundation of China (No. 31772450 and 31600308) and the Project of Service to Industrial Innovation of Higher Education, Shanxi province: Discipline Group of Ecological Remediation of Soil Pollution. The funders had no role in study design, data collection and analysis, decision to publish, or preparation of the manuscript.

### Grant Disclosures

The following grant information was disclosed by the authors:
National Natural Science Foundation of China: 31772450, 31600308.
Project of Service to Industrial Innovation of Higher Education, Shanxi province: Discipline Group of Ecological Remediation of Soil Pollution.

### Competing Interests

The authors declare there are no competing interests.

## Author Contributions

- Pengyu Zhao conceived and designed the experiments, performed the experiments, analyzed the data, contributed reagents/materials/analysis tools, prepared figures and/or tables, authored or reviewed drafts of the paper, approved the final draft.
- Jiabing Bao, Xue Wang and Yi Liu performed the experiments, approved the final draft.
- Cui Li performed the experiments, analyzed the data, approved the final draft.
- Baofeng Chai conceived and designed the experiments, authored or reviewed drafts of the paper, approved the final draft.

## Data Availability

The raw data are available in the Supplementary Files. The bacterial sequences have been deposited in the SRA database: SRP135838.

## Supplemental Information

Supplemental information for this article can be found online at http://dx.doi.org/10.7717/peerj.6746#supplemental-information.

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
