# Peer review of "Deterministic processes dominate soil microbial community assembly in subalpine coniferous forests on the Loess Plateau"

_PeerJ, doi:10.7717/peerj.6746_

## Round 0.1 · original submission · Major Revisions

Dear authors,

As you will see, both reviewers found your work of interest, but they have also identified major issues that need to be addressed to make the ms. acceptable for publication.

From my side, I want to stress the following: better account for previous work published in this domain, being more humble on the novelty of the work when needed ; focus the ms. on more precise questions deepening your analyses to tackle these ; complete information in the Mat&Met section, including for replication ; avoid excessive speculation. More generally, try not to jumble together many analyses using many packages available. Rather, focus your ms. following a clearer logic.
Please carefully take into account all the constructive comments and suggestions made by both reviewers (including an annotated pdf from Reviewer 2).

Looking forward to receiving a revised version.

Best regards
Xavier LE ROUX

Reviewer 1 ·

Basic reporting

Clear English used but lot of jargon. The writing style is ambiguous, where many broad level statements are made but not backed up with data or relevant references.
Literature references are not thorough to provide sufficient background information.
Raw data has not been shared, neither code for the models.
There is extensive speculation with the second and third hypotheses.

Experimental design

No justification provided on why the sites were chosen. Neither are replicates highlighted. How were the 23 samples divided between the three sites? How were the replicates handled?
No clear statement on how knowledge gaps were filled.
Methods are not described in sufficient detail, and definitely no information to replicate.

Validity of the findings

Unclear how replication was handled, which means statistical soundness of the data cannot be evaluated. Conclusion is not well stated, and there is broad speculation.

Additional comments

Line Specific Comments:
Line 65: Check format of references
Line 69, Line 75: What does one-sidedness of analytical methods mean? The authors repeatedly use this phrase but have not provided any explanation of what this actually means? What is this sidedness? What is the other-side that needs to be analyzed?
Line 80: What exactly do the authors mean by precisely and robustly disentangle different ecological processes? In general, there are many such sentences throughout the manuscript that do not really explain the point the authors are trying to make.
Line 81: For readers not familiar, what do NRI and NTI actually mean? This should be highlighted here as opposed to Lines 147-148.
Line 86: Aggregation of what processes?
Line 88-89: Measured traits usually represent the functioning of an organism. It is a grey area when the authors write that traits cannot represent the ‘whole’ functioning. What does one mean by whole functioning of an organism? Can the authors provide examples where functional traits have failed to represent the functioning of an organism?
Line 106: How many plots each in WT, LY, and PGQ? Why these broad mountain distinctions? The authors don’t mention this until very later in the manuscript. Why were these sites chosen? How many replicates per plot? Are all results presented downstream of replicate samples, pooled samples? Statistics were performed on what? It is hard to extract this information downstream when the only information provided is 23 samples were studied.
Line 111: 1 g of sieved soil was used for DNA extraction? In that case, the extraction does not represent actual community profile. Soil sieving significantly impacts microbial community structure.
Line 145: Provide reference for Stegen et al.
Line 137: How many times (iterations) was the model run?
Line 158: Show how one arrives at niche breadth from OTU table?
Line 167-168: Where is this functional niche breadth-based dendogram? Is this something the authors created or is this something widely available?
Line 198: Ammonium nitrogen can’t be significantly related to itself.
Lines 203-216: Why should the readers care about this? How are the sites different to begin with? Should a difference be expected?
Line 221: Looks like pH has the longest length
Lines 239-240: It is unclear how the authors arrived at the conclusion that changes of modules represented changes in interspecific relationships. The network analysis figures are impossible to decipher. What do the colors mean? The networks look very tightly linked. What do the modules mean? How many modules were there?
Line 258-260: There is no substantial evidence presented in the manuscript to back this.
Line 261: What one-sidedness?
Line 264-265: Where is the proof that interspecific relationships are driving community assembly? What are these relationships? Are these selection, dispersal ?
Line 280: It will make most sense to regress the environmental variables with results of assembly processes to avoid speculations. If indeed SOC Is the main driving force, this should be visible in the regression with assembly process.
Line 295-302: This should go in the introduction as an added information about the sites.
Line 303-304: Again, this is a stretch to say interspecific relationships (what are these, what is the metric for this?) explained the variation.
Line 321-324: Highly speculative. Litter dependent nutrient accumulation and therefore substrate availability can be tested. Was competition accelerated at one site, across all sites, between sites?
Line 325-340: Applying macroecology principles of deep and shallow phylogeny related historical factors to microecology is questionable. Especially in microbes where horizontal gene transfer is a very active process, the authors have not presented any proof that such extrapolation from macro to micro ecology is valid. This paragraph sheds no extra light, and ends up confusing the reader. What is meant by disjunctive occurrences?
Line 338-340: Did the authors validate that there is no different in paleo and current climate or the difference was not the driving factor?
Lines 341-348: The purpose of this paragraph fails me. What are the authors trying to say?
Line 350-352: The authors should have just stopped at this conclusion. This is the only definitive outcome of this study. Rest is highly speculative and not clearly written, proven, or discussed, and there is use of jargon. For readers less familiar with the field of study, this manuscript does not provide clarity on biogeography of soil bacterial communities, if that is what the authors were aiming for.
Line 361: Where are the source codes for all the modeling? Have these been deposited in any database? If someone attempted to repeat this experiment/analysis, how would they do so? This cannot be repeated.
Lines 366-520: References have to be heavily formatted. Many inconsistencies.

·

Basic reporting

The study is about an environmental gradient in a sub-alpine forest in China using 23 plots that they clump in 3 categories based on the mountain ranges. They use 16S tags in illumina and the nutrient content of the soil. The manuscript starts very promising and their hold of community theory is in general sound. However, their review of the recent papers on the issue is not so sound. This is not the first community analysis of soils that has attempted to join several points of view in one analysis (google scholar has14,000 in the last 4 years, many of them with a very deep understanding of the problem).
I particularly like: Delgado-Baquerizo et al., 2016. Microbial diversity drives multifunctionality in terrestrial ecosystems. Nature Communications. http://dx.doi.org/10.1038/ncomms10541. The authors do not know the literature so they try to "discover" new methods, without really going deep into any of them. The authors look for differences between stochastic vs deterministic drivers by looking at the relationship between environmental variables, phylogenetic relatedness and interactions using network analysis (that they do not describe). However, even if they have more than 4000 different OTUs most of the analysis seems to be done at the phyla level or class level losing most of its value, since patterns with such a rough “grain” are unable to disentangle the effect of, precisely, interactions. As an example is the figure 8 where there is nothing to be seen in the “clouds”.

Experimental design

The basic design is good, the number of samples and the sampling is well done, even if always more sequences can be obtained, that is not the point. The theory is the correct one, even if they need to read more and be humble about their uniqueness. The problem here is the analysis, this is why this can be improved if they read and are careful about their statistical approach. They need to explain better at what level is each test, and disentangle the patterns that are jumbled together. The devil is in the details and here there are a lot of details to sharpen. Figure 7 does not make sense, at its presence is the core of the problem, 78% is an unknown…probably due to the resolution of the analysis done, or the lack of the proper question to the analysis. It seems that somehow, they did all the potential analysis, put them in a blender and got lost in the way. Please rescue the “tread” of the story.

Validity of the findings

The only clear pattern is the relationship with nutrients (RDA analysis). One mountain range is more diverse than the other…but the difference between stochastic vs deterministic drivers that explain such diversity is not clear at all. There is a lot of polish that needs to be done in order to add value to this simple observations.
I recommend that you follow the network analysis done by De Anda et al., 2018 in frontiers, it is an example of a careful network analysis.

Additional comments

I am putting mayor revision given the potential of a good paper if the proper analysis is done. I am adding the pdf with my direct comments, do not pay attention of the highlighting, that was for me to follow the story.

---

## Round 0.2 · Major Revisions

Dear Pengyu,
Your manuscript has been re-reviewed by two external reviewers. One reviewer (R1) indicated that the revised ms. has been much improved, but the other reviewer found that the ms. still suffers from deficiencies.
Could you please carefully address all the points raised, in particular:
(1) Be clearer on the hypothesis / justification for conducting the study. For instance, did you aim at relating community assembly patterns to an elevation gradient?, or to a chrono/climate-sequence?, or to a soil/ecosystem successional stage? Provide a clear rationale, with information on what the climate&soils were like?
(2) Based on (1), better detail which continuum was expected in the sub-alpine ecosystem?
(3) Explain why the number of samples collected differs between the sites, and to what extent this can be a problem or not when analysing the data (be specific, with clear rationale).
(4) Test bNTI against SOC matrix using a mantel test.
(5) Cite relevant references like Jurburg S., et al. 2017. Sci. Reports 7: 45691 DOI: 10.1038/srep45691
(6) More generally, take into account the detailed suggestions made by the first reviewer.

I look forward to receiving a re-revised version of your manuscript along with a point by point response to the reviewers' comments and suggestions.

best regards
Xavier

Reviewer 1 ·

Basic reporting

1. No clear hypothesis or justification for conducting the study.
2. What continuum was expected in the sub-alpine ecosystem?
3. The sampling design is flawed (with varying number of samples collected from the sites). Extremely difficult to infer any meaningful conclusion from the statistics.
4. With no conclusive reasoning for the study, there is no understanding of what knowledge gap was identified and addressed.

Experimental design

Flawed. 10 samples were collected from one site, 8 from another, and 5 from the third. No information provided in the methods as to how this was dealt with/how was normalization done. There is inherent heterogeneity within each site as well. Research question in not well defined or meaningful.
Moreover bNTI needs to be tested against SOC matrix using a mantel test.

Validity of the findings

With no hypothesis to test, and irregular sampling strategy, no justification of why samples were studied in sub-alpine system (e.g. were the authors aiming for change in elevation gradient and community assembly patterns?, or if the authors were attempting to study a chrono/climo-sequence, how were their sites classified, or if the authors were attempting to study soil/ecosystem successional stage, what were their soils like?

Additional comments

Line 26: Check sentence structure
Line 33: Abrupt sentence
Line 28-29: What is this continuum? What are the boundaries of this? What was more prevalent at which site?
Line 34: Recommended to perform a mantel test between bNTI and SOC matrices to understand if SOC is really driving the assembly process. The SOC numbers vary considerably within each site, and integrated RDA, MRT, and correlation analysis was not performed on the community assembly results. This is a indirect inference, which can be made stronger and conclusive if the mantel test is done.
Line 35: what are these ‘some’ differences? The abstract needs to be have conclusive results and not general pronouns/adverbs to describe the results.
Line 36-37: The abstract does not provide any information of this continuum, how is the continuum being defined? Did one see stochastic processes at one elevation, which gradually change to deterministic as one moved higher in elevation? Or did the assembly structure change progressively from stochastic to deterministic as SOC increased? The word continuum has been repeated a few times in the abstract, but it is not clear as to what this continuum is.
Line 48-49: Please cite Baas-Becking for the “Everything is everywhere…” sentence.
Line 58: What is the authors’ interpretation of this continuum? Or what is the accepted interpretation of this continuum in the field of microbial ecology?
Line 62: Based on these previous studies where ecological continuum varies based on environmental conditions, can the authors derive hypothesis for their own study?
Line 66-68: What common standard? The studies are ‘limited’ to specific spatial and temporal scales or sampling scales because the question being asked in those studies are probably relevant at those scales. It is unclear what common standard the authors are referring to and trying to prove?
Line 69: How does sampling sub-alpine coniferous forest support the ‘common standard’ that the authors are trying to achieve?
Line 77-78: The sites are not along a transect of elevation.
Line 79: What are the climatic gradients observed for the three sites? Why should a difference be expected between the sites enough to warrant a community assembly study?
Line 83-87: What is the justification for such random sampling (8, 10, 5) from each site? The sites itself are heterogeneous. For the WT site, with such a range in the elevation, where was the sampling done and why? Can’t the authors group their samples according to elevation? There seems to be no logic evident for the sampling strategy. Moreover, if each plot is treated as a replicate for the site (which shouldn’t be the case since the sites are heterogeneous to begin with), then it appears that there were 8 replicates in one site, 10 in another, and 5 in the last one. It is extremely difficult to perform statistics and infer meaningful results with this sampling design.
Line 104: If you are using primers for prokaryotic 16S rRNA gene, there shouldn’t be any eukaryotic sequences to remove.
Line 119: Reference?
Line 121: How was the model controlled for stochastic variation and associated changes in alpha diversity?
Line 163-164: which analysis did you use? Generally people use one or the other.
Line 164-165: Why not spearman correlation analysis between otus and environmental variables?
Line 165: What is PCNM?
Line 166: What are these 23 extracted PCNM variables?
Line 184-186: Define the gradient. These lines finally somewhat depict the reason for conducting community assembly studies in the sites that were selected. This should be reflected in the introduction.
Line 192: Enough for what?
Line 220-221: I don’t think anything can be commented about deterministic vs stochastic responses from partitioning analysis.
Line 225: Network Analysis: PQG site has maximum sampling sites (10). It is not surprising that PQG exhibited more edges, vertices, degree and clustering coefficients. The analysis is biased because the sample size is uneven across the sites. How was this factored for in this study?
Line 227: Fewer modular?
Line 230-231: Nothing can be inferred about keystone species at different sites since the sampling size was different and therefore the results are not comparable.
Line 240: Which parameters of the assembly process?
Line 247: How does one manage microorganisms? There are billions of them in 1 g of soil.
Line 251-252: This study focused on bacteria. How does a comparison to fungal results make any sense here?
Line 254: Determination?
Line 262: Compared to this literature, does this study actually capture early vs late successional soils? What is the importance of the sites? What gradient was being studied, if any?
Line 289: The irregularity in the number of plots sampled per site makes this analysis very hard to interpret.
Line 300: Makes no sense
Line 307: “some” differences?
Line 307-309: Unfortunately I did not find any extra knowledge from this study, or what this continuum is, and the study contributed to understanding the continuum.
Line 313-320: Check these references, they seem out of place and not arranged alphabetically.
Figure 3a, b, c: How were the individual samples handled in these plots? Were the 8, 10, and 5 samples from the sites averaged per site?
Figure 6a: The legend colors cannot be differentiated. The figure is hard to interpret.
Figure 6b: Write in the text that spearman correlation was performed.
Figure 9: bNTI is a matrix. You cannot do ANOVA to test relation with SOC. You need to do mantel test. Figure 9C and 9D is supposed to show R2 values. Not sure what R is.

·

Basic reporting

This is a resubmission of a previously evaluated document, the authors did all the suggestions of the reviewers and I consider that the manuscript is mostly ready for acceptance. The english is much better and the the speculation about the results have notable been tone down.
This is an interesting study where they survey an altitude gradient for soil microbiome observing that carbon is the main driver of diversity, confirming other studies.

Experimental design

It is a good design with enough replicates and samples for 23 sites

Validity of the findings

It is well done

Additional comments

I believe that this version of the manuscript is good for publication.

---

## Round 0.3 · Major Revisions

Dear Authors,

One of the reviewer has evaluated this revised version of the ms.
While you have improved the ms., you have still not made clear what was your main research question and hypothesis.

I agree with the reviewer. Actually, I think that you are still not clear on what your experimental design is, allowing to answer which question with which hypothesis. it seems you have to be clear which of the 3 following options are the relevant one:

(1) you compare 3 sites having different characteristics and expect assemblage processes to vary among them.
(2) you analyse how assemblage processes vary along 3 environmental gradients (one per site)
(3) you analyse how assemblage processes vary along an environmental gradient (from lower to upper distribution zone) replicated 3 times (3 sites)

Whatever the option, you should make it clear in the ms. (including in the title, abstract and introduction) and then state why and what was your main hypothesis.

I am ready to consider a last revised version to evaluate whether you can make this very clear throughout the ms. Would you not be in a position to make this clear enough, I would have to reject your paper.

I had a quick look at you ms. to indicate the key parts of the title/abstract/introduction/M&M which should make this very clear. Then you will have to ensure the results and Discussion sections are written according to the choice you will make.

Looking forward to reading the revised version

best regards

Xavier

Reviewer 1 ·

Basic reporting

Improved version from the last time.

Experimental design

Still a disjoint between what the hypothesis is and how the sampling is suitably set-up to test the hypothesis. Please see line specific comments.

Validity of the findings

Conclusions are not well stated. See line specific comments.

Additional comments

General comments: I still do not see a compelling reasoning for this study or a clear well-defined hypothesis. It is not suffice to say soil A is different from soil B. Given how much heterogeneity exists in soil environments, ofcourse soil A will be different from soil B. The justification for this study is still sending out mixed signals, as if the authors do not have a clear idea of what they are attempting to study and why. Sometimes it reads as if they are studying altitudinal gradient, sometimes it reads that they are studying a climatic gradient and climosequence, and it also reads as if they are studying assembly processes that shift across the lower and upper distribution limit of a subalpine tree species. What is it that is exactly being studied here and why? If the upper and lower limit of distribution define the sampling sites, are the authors studying community assembly in the soils at these zones of distribution? Is that related to the ecology of the trees? Perhaps higher litter at higher distribution areas indicate higher SOC, and therefore a more deterministic force?

Line Specific Comments:
Line 70: Why would the relative importance differ? Were your sites representing successional stages?
Line 73: “different scales” of what?
Line 73-74: I do not see any theoretical support being provided to understand the mechanism. Even after reading the entire manuscript, I do not see a link between the environment that was studied and the results that were obtained. Without a strong hypothesis, the reason for this study is lacking.

Line 82: What is meant by typical soil plots?
Line 84-86: The sentence sounds like these references studied the same site as yours and provided conclusions about microbial community in your sites. This is factually incorrect.

Line 87: Capitalize Cinnamon. This is a soil series.

Line 88-92: The driver gradients are made up of what? Still sending out mixed signals, sometimes it reads as if you are studying altitudinal gradient, sometimes it reads that you are studying a climatic gradient and climosequence, and it also reads as if you are studying assembly processes that shift across the lower and upper distribution limit of a tree. What is it that is exactly being studied here and why? If the upper and lower limit of distribution define the sampling sites, are you studying community assembly in the soils at these zones of distribution? Is that related to the ecology of the trees?

Line 93-98: Which site was at the lower distribution limit and which site was at the upper distribution limit?

Line 104: Sieving significantly alters microbial community composition. Justify why this was done. How are you confident that this is representative of the actual microbial community composition at the site?
Line 117: Incorrect reference
Line 118: Why was archaea removed?
Line 141: Incorrect sentence
Line 241: modular what?
Line 253-254: In your response document you said that bNTI values are not matrix. How did you do mantel test (which you claim to have done) if the bNTI values are not a matrix? Mantel test is for matrices.
Line 258-259: Weird sentence
Line 262-264: Again, you do not have fungal community data here. It makes no logical sense to therefore compare your bacterial community assembly work to someone else’s fungal community work.
Line 271-274: Do the soils in this study fall under any of these category described here?
Line 285-290: Have you considered if SOC is impacted by the upper and lower distribution of your tree species. Litter from the trees will impact SOC, which in turn will impact the community assembly structure, and is perhaps a reason that you see variable selection increase from LY to WT sites.
Line 290-292: Are your sites representative of successional stages?
Line 302-304: Perhaps brought about by transition zones between LY and WT? Why is there greater heterogeneity at PQG?
Line 315-321: Not a strong conclusion. Needs to be refined. What inference can one draw about subalpline coniferous forest density vs soil microbial community assembly processes?
Figure 3: Please write in the caption that these represent average values.
Please check significant figures in Table S1 pH values.
Why is Oikos et al.’s supplementary appendix material being provided? The codes are for that study. How is it relevant to this current study? I am not even sure one can just provide another study’s appendix in one’s supplemental material.

---

## Round 0.4 · Minor Revisions

Dear authors,

the revised version will be acceptable for publication, provided you adjust the text slightly (see suggested adjustments in the attached file - staff will also send you the original Word doc)

best regards
Xavier LE ROUX

---

## Round 0.5 · accepted · Accept

Thank you for the final polishing of your ms. which is now acceptable for publication in PeerJ.

Best regards
Xavier LE ROUX

#